# ROBOTIC STEERING: MECHANISTIC FINETUNING OF VISION-LANGUAGE-ACTION MODELS

## ABSTRACT

Vision-Language Action Models (VLAs) promise to extend the remarkable success of foundation models in vision and language to robotics. Yet, unlike those models, usable VLAs for robotics require finetuning to contend with complex physical factors like robot embodiment, environment characteristics, and spatial relationships. Current fine-tuning methods adapt the same set of parameters regardless of the visual, linguistic, and physical characteristics of a particular task. Inspired by functional specificity in neuroscience, we hypothesize that it is *more effective to fine-tune components of model representations specific to a given task.* In this work, we introduce **Robotic Steering**, a novel mechanistic finetuning approach that identifies task-specific representations in the attention-head space to selectively adapt VLAs. In particular, we use few-shot examples to identify and selectively finetune only the VLA attention heads that align with the specific physical, visual, and linguistic requirements of a task. Through comprehensive on-robot evaluations using a Franka Emika robot arm, we demonstrate that Robotic Steering matches or outperforms full-head LoRA across all tested tasks. Crucially, Robotic Steering demonstrates superior robustness under environmental and task variations compared to standard LoRA finetuning, while enabling faster, more compute-efficient, and interpretable experimentation. Grounded in mechanistic interpretability, Robotic Steering offers a controllable, efficient, and generalizable framework for adapting VLAs to the diverse physical requirements of robot tasks.

## 1 INTRODUCTION

*"It is tempting, if the only tool you have is a hammer, to treat everything as if it were a nail."*

— Abrahan Harold Maslow, *The Psychology of Science* [46]

Vision-Language-Action (VLA) models represent an emerging paradigm that extends foundation models to robotics by applying next token prediction across vision, language, and physical action spaces [37, 49, 62, 70]. While large-scale robotic datasets [12, 14, 36] have enabled unprecedented training scales, VLAs have yet to achieve the impressive generalization of language and vision-language models. Unlike those models that demonstrate remarkable zero-shot adaptation, VLAs require targeted finetuning for each specific deployment environment, establishing a paradigm where practitioners must adapt models to match the exact specifications of their intended task.

This reality raises a philosophical question: what constitutes a "task" in robotics? A seemingly straightforward manipulation objective such as picking up a mug can have many physical instantiations when considering real-world perturbations [2, 22] such as a camera position, the color of the mug, the table height, or even variations of the robot initial position by a few centimeters. Unlike vision and language domains where tasks have clear boundaries, robotics operates in a continuous space of physical variations where the slightest environmental perturbation can fundamentally alter the required model behavior. We propose that few-shot expert demonstrations better specify what a robotic "task" is, as they contain the valuable physical information inextricably linked to the task definition. Unlike linguistic descriptions alone, these demonstrations encode the physical properties

Figure 1: **Robotic Steering enables efficient task adaptation** by finetuning specific heads of a vision-language-action (VLA) model. Standard finetuning trains all parameters, and by intelligently selecting heads to finetune, we find Robotic Steering to be more interpretable and robust to distractors.

of the deployment scenario: the exact angle the robot grasps from, how cluttered the workspace is, what lighting conditions exist, and countless other factors that determine successful execution.

Given task specification through few-shot demonstrations, the key challenge becomes: how can we effectively make use of these demonstrations to learn an embodied task efficiently? Current finetuning methods like LoRA [31] adapt the *same set of parameters regardless of the specific requirements of each task*. In contrast, we take inspiration from functional specificity in neuroscience, which suggests that certain brain regions are specialized for particular tasks [19, 35], and from mechanistic interpretability in machine learning, which has shown that specific attention heads in transformers encode distinct capabilities [27, 50]. Building on these insights, we introduce a novel paradigm: using the few-shot demonstrations themselves to identify which attention heads encode the task-relevant representations, then selectively finetuning only those components. This approach recognizes that different tasks engage different model capabilities, for example grasping from above requires different visual and spatial reasoning than pushing sideways, and adapts the model accordingly.

We introduce Robotic Steering, the first approach to leverage mechanistic interpretability for fine-tuning task-specific representations of VLAs. Our method consists of three steps, each addressing a key challenge in VLA adaptation. First, we perform semantic attribution to identify task-relevant attention heads. Given a set of few-shot demonstrations of a task, we extract activations from each attention head as the base model performs a forward pass on the examples. We then select heads whose activations perform best on a lightweight k-NN regression task to predict the ground truth actions for the examples. By identifying these task-specific heads, we can achieve more precise adaptation than uniformly finetuning all parameters. Our second step is to freeze the visual encoder, action expert, and LLM backbone while applying targeted finetuning to only the queries and MLP parameters associated with selected heads using LoRA adaptors. Finally, the resulting model deploys as a standard checkpoint without additional overhead. Unlike other mechanistic approaches that require activation interventions during runtime, our finetuned weights integrate seamlessly into existing VLA deployment pipelines. An overview is shown in Figure 1 and Figure 2.

We summarize the main contributions of our work: (i) We introduce Robotic Steering, the first method combining mechanistic interpretability with robotic finetuning for controllable adaptation through semantic attribution of attention heads; (ii) Through comprehensive on-robot evaluations using a Franka Emika robot arm, we demonstrate that Robotic Steering matches or outperforms full-head LoRA across all tested tasks while requiring less runtime and fewer parameters; (iii) Our approach exhibits superior generalization under environmental distractors, including variations in lighting, object properties, and scene configurations, compared to standard finetuning methods; (iv) We provide a practical framework producing standard model checkpoints deployable without additional inference overhead, making mechanistic finetuning accessible for real-world robotic systems.

## 2 RELATED WORK

**Few-Shot Adaptation in Vision-Language-Action Models**. Large Language Models (LLMs) [3, 34, 54, 64] and Large Multimodal Models (LMMs) [1, 4, 42, 43, 51, 60, 61] have demonstrated

remarkable capabilities through large-scale pretraining and causal token prediction. Vision-Language-Action models (VLAs) represent the current frontier of robot policy learning [15, 37, 55, 62, 70] and is enabled by large-scale datasets [12, 14, 36]. This scale of training has demonstrated generalization across embodiments and tasks. The state-of-the-art $\pi$-series models—$\pi_0$ [10] and $\pi_{0.5}$ [53]—use flow matching for continuous action generation along with large-scale data to achieve impressive zero-shot transfer. Despite these advances, VLAs struggle with few-shot adaptation to new environments.

Researchers have explored various few-shot techniques: in-context learning approaches [45, 58, 67] condition on demonstrations without weight updates but face context limitations; parameter-efficient methods [25, 31, 32, 39, 44] and specialized adaptations [38, 57] reduce trainable parameters; meta-learning [20, 23, 68] and behavior retrieval [17, 40, 66] enable rapid adaptation given access to prior data. However, these methods operate at the level of entire weight matrices without considering which components encode physical reasoning. Thus, they lack interpretability and fail to leverage VLAs' structured representations, motivating our mechanistic approach. We also note other work in steering in robotics focuses on guiding the action denoising process of diffusion policies [16, 48] or designing inference-time action sampling metrics [48]. Instead, our work takes a mechanistic approach that more selectively finetunes a VLA.

**Mechanistic Interpretability**. Recent advances in mechanistic interpretability have revealed how model behavior can be precisely manipulated through internal representations. Early research [8, 9, 69] established frameworks for understanding semantic encoding in neural networks, while activation steering methods [52, 59, 65] demonstrated parameter-free behavior modification. The discovery of specialized components like induction heads [50] and task-specific neurons [28] led to task vector abstractions [26, 63], with parallel work on sparse autoencoders [13] and superposition [18] providing tools for decomposing representations.

An emerging line of work leverages few-shot mechanistic interpretability for model adaptation through task vector methods [11, 29, 33, 47], which concentrate task-relevant information in specific attention heads or activation subspaces for efficient parameter-free adaptation. Research in multimodal representations has revealed how vision-language models structure cross-modal concepts through multimodal neurons [24], mechanistic understanding [56], text-based decomposition [5, 21], and knowledge localization [6, 7]. The comprehensive survey by Lin et al. [41] provides a broader overview of these approaches. While these methods have succeeded in language and vision domains, our work is the first to apply mechanistic interpretability to vision-language-action models, leveraging these insights to identify and adapt components responsible for physical reasoning in robotic control.

## 3 METHODS

In this section, we present Robotic Steering, our approach for enabling finetuning of task-specific components of Vision-Language-Action models through mechanistic interpretability. Our method identifies and selectively finetunes attention heads that encode task-relevant physical reasoning, allowing VLAs to learn new capabilities while preserving existing ones. We begin with preliminaries on VLA architectures, followed by our three-step approach: (1) identifying task-relevant attention heads through k-NN regression, (2) selective finetuning of identified components, and (3) standard inference with finetuned weights.

### 3.1 PRELIMINARIES

**Vision-Language-Action Models**. VLAs extend the transformer architecture to robotic control by processing visual observations and language instructions to predict continuous action vectors. Given an observation $o_t$ consisting of image frames and optional language instruction, a VLA predicts an action vector $a_t \in \mathbb{R}^d$ containing control values (e.g., joint velocities, gripper commands). Modern VLAs like $\pi_0$ [10] formulate this as a conditional generation problem, where actions are produced through autoregressive token prediction or flow matching. The model processes inputs as a sequence of visual tokens, language tokens, and robot state information, combining multimodal information for action prediction.

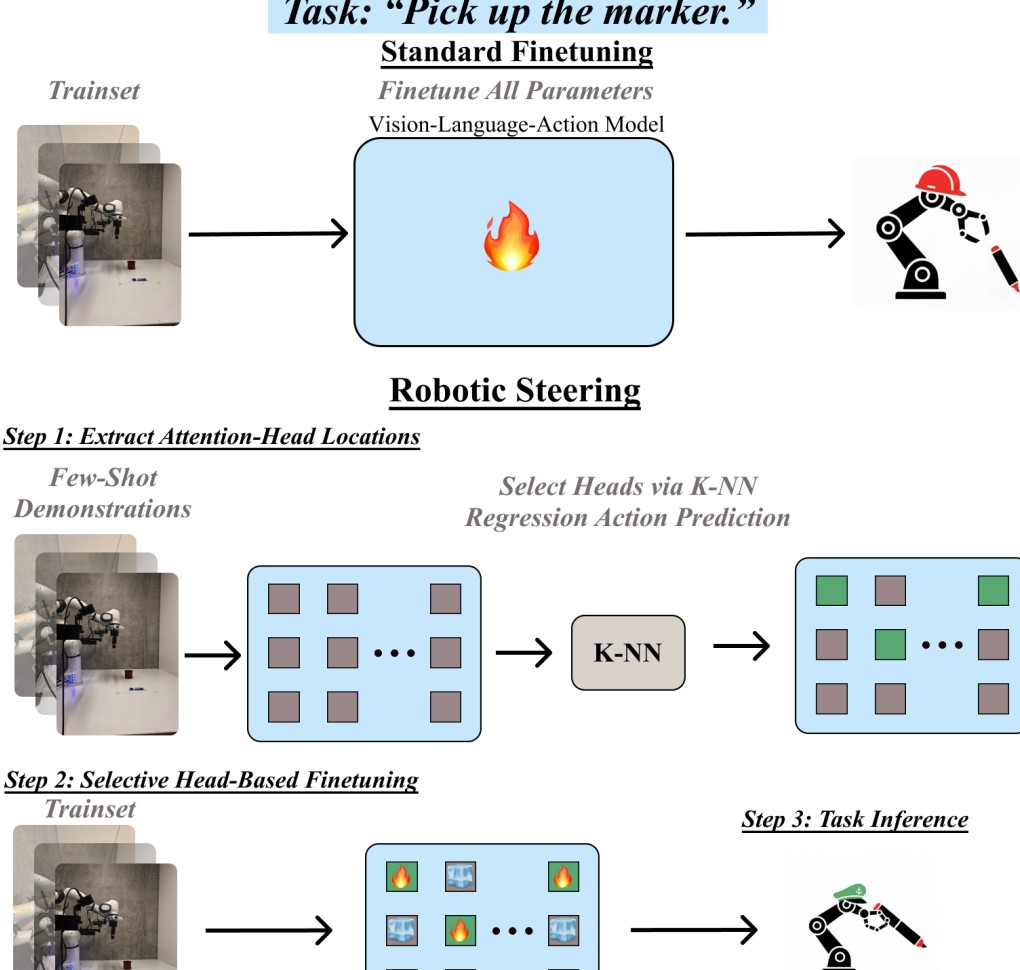

Figure 2: **Method.** Robotic Steering enables targeted adaptation of VLAs by detecting attention heads encoding task-relevant information, finetuning only these components, and reusing the updated model for standard inference.

**Multi-Head Attention**. For a transformer with $L$ layers and $H$ attention heads per layer, each head $(l, h)$ computes:

$$\mathbf{h}_l^h(x_i) = \text{softmax}\left(\frac{QK^T}{\sqrt{d_h}}\right) V \tag{1}$$

where $Q$, $K$, $V$ are query, key, and value projections. For action prediction in VLAs, we focus on activations at the final token position $\mathbf{h}_l^h(x_T)$, which aggregates information across the entire sequence.

### 3.2 STEP 1: IDENTIFYING TASK-RELEVANT ATTENTION HEADS

Our key insight is that within a VLA's attention mechanism, specific heads naturally specialize in encoding physical concepts relevant to particular manipulation tasks. We identify these heads through their ability to retrieve examples with similar action patterns.

**Extracting Head Activations**. Suppose we are given a frozen VLA and few-shot demonstrations $\mathcal{D} = \{(\tau_1, a_1), (\tau_2, a_2), \ldots, (\tau_N, a_N)\}$, where each trajectory $\tau_i$ consists of $T$ timesteps. Each timestep $t$ contains the VLA's input observation: visual tokens from camera images, language tokens

from task instructions, and robot state information (e.g., joint angles). The corresponding $a_i \in \mathbb{R}^{T \times d}$ are action vectors across all timesteps.

For each timestep $t$ in trajectory $\tau_i$, we extract the attention vector $\mathbf{h}_l^h(\tau_i^t)$ for every head $(l, h)$. Importantly, we work at the timestep level rather than trajectory level—each timestep becomes an individual example in our retrieval set.

**k-NN Regression for Head Evaluation**. To evaluate each head's relevance, we assess its ability to retrieve timesteps with similar actions. The intuition is that if a head's representation groups together observations that require similar physical actions, then this head encodes task-relevant features worth finetuning. In order to make head selection more efficient, we employ the keyframe extraction approach suggested in [67]. Functionally, however, the approach is identical with or without this step. More details can be found in Section A.2 of the Supplementary material.

For a query observation $q$ from trajectory $\tau_i$ at timestep $t$:

We first find the $k$ nearest neighbor timesteps from all other trajectories based on cosine similarity in head $(l, h)$'s representation space:

$$\mathcal{N}_k^{l,h}(q) = \text{top-}k \left\{ \frac{\mathbf{h}_l^h(q) \cdot \mathbf{h}_l^h(\tau_j^s)}{\|\mathbf{h}_l^h(q)\| \|\mathbf{h}_l^h(\tau_j^s)\|} \right\}_{j \neq i, s} \tag{2}$$

Second, we predict the action by averaging the actions of retrieved neighbors:

$$\hat{a}_t^{l,h} = \frac{1}{k} \sum_{(\tau_j^s) \in \mathcal{N}_k^{l,h}(q)} a_j^s \tag{3}$$

Finally, we compute the head's score as the mean squared error across all queries:

$$\text{score}(l, h) = \frac{1}{|\mathcal{D}| \cdot T} \sum_{\tau_i \in \mathcal{D}} \sum_{t=1}^{T} \|\hat{a}_t^{l,h} - a_i^t\|^2 \tag{4}$$

We select the top-$m$ heads with lowest scores:

$$\mathcal{H}_{\text{task}} = \{(l, h) \mid \text{score}(l, h) \text{ is among } m \text{ lowest scores}\} \tag{5}$$

These heads learn representations that effectively map task-specific observations to other observations requiring similar actions within the few-shot demonstration trajectories, making them ideal candidates for task-specific finetuning.

### 3.3 STEP 2: SELECTIVE FINETUNING WITH LORA

Having identified task-relevant heads $\mathcal{H}_{\text{task}}$, we perform targeted finetuning while preserving the model's general capabilities.

**Sparse Parameter Updates**. We freeze all model components except the query projections of selected heads. For each head $(l, h) \in \mathcal{H}_{\text{task}}$, we apply Low-Rank Adaptation (LoRA) [31]:

$$W_Q'^{l,h} = W_Q^{l,h} + B^{l,h} A^{l,h} \tag{6}$$

where $B^{l,h} \in \mathbb{R}^{d \times r}$ and $A^{l,h} \in \mathbb{R}^{r \times d}$ are low-rank matrices with rank $r \ll d$. We also finetune the MLP layers associated with the selected attention blocks.

**Training Objective**. Our approach is flexible and compatible with any VLA training objective. We simply finetune the selected heads using the same loss function as the base model—whether that's flow matching loss for diffusion-based models like $\pi_0$ or cross-entropy for discretized action spaces. This selective updating acts as a targeted refinement that enhances task performance without broadly overwriting the model's parameters.

### 3.4 STEP 3: INFERENCE

After selective finetuning, inference proceeds through standard forward passes with the finetuned weights. Unlike many mechanistic interpretability methods that require computing and manipulating

Table 1: Performance comparison of Robotic Steering on new and in-domain tasks. Methods are finetuned on tasks with varying example sizes (20-200), then evaluated on both new and original in-domain tasks for 20 trials under the same task and environmental settings.

| Method | Training Time | Trainable Params | New Tasks | | In-Domain Tasks | | |
| --- | --- | --- | --- | --- | --- | --- | --- |
| | | | Place Marker in Cup | Push Button Hard | Pick Cube | Place Cube in Bowl | Push Bowl to Cup |
| Zero-shot | - | - | 10% | 0% | 0% | 0% | 0% |
| Full-head LoRA | 239 min | 1785.9M | 75% | 65% | 75% | 60% | 60% |
| **Robotic Steering (KNN)** | 189 min | 78.8M | **80%** | **75%** | **90%** | **85%** | **65%** |

activations at inference time, our approach produces a standard model checkpoint deployable without additional computational overhead or specialized procedures. The model simply uses the finetuned weights for the selected heads while maintaining frozen weights elsewhere, preserving both new task capabilities and existing skills through this selective modification.

# 4 EVALUATION

In our work, we evaluate our method on a variety of real-world on-robot tasks using the strong $\pi_0$ VLA to demonstrate the effectiveness of our approach on realistic, physically-grounded usecases. We select tasks of diverse difficulties and skills and deeper experimentation and ablation that showcases the many unique qualities of our approach including its performance, robustness, and interpretability. We present more details as follows:

## 4.1 IMPLEMENTATION DETAILS

While our method is model-agnostic, we use $\pi_0$ [10], a state-of-the-art VLA that uses flow matching for continuous action generation. Our entire implementation is in Jax [], which notably lacks convenient hooks to easily extract activations from the model. Thus, we highlight the development of such functionality for a Jax-based model as a core technical contribution of our work. We finetune the model using 2 NVIDIA RTX A6000 GPUs, emphasizing the lightweight nature of our approach. We extract attention activations from the model's PaliGemma [] LLM backbone with 18 layers with 8 heads each, selecting $m = 20$ heads for finetuning based on k-NN regression with $k = 5$ neighbors. The LoRA rank is set to $r = 8$, and we finetune for 5,000 steps for our main experiments using varying number of demonstrations depending on the difficulty of the task. More implementation details can be found in Supplementary Section B.

## 4.2 ROBOTIC SETUP DETAILS

We follow the setup from DROID [36] exactly, using a 7-DoF Franka Emika Panda robot arm with a Robotiq gripper and a low-level Polymetis controller []. As suggested by DROID, we enable two of the three cameras for both finetuning and inference: the left arm camera and wrist camera. We record each example episode at 6 Hz. All data collection is performed on-robot using teleoperation, with each task controlling for the exact objects used to ensure fair evaluation across methods.

We evaluate a total of 5 primary tasks with the following language instructions: (1) "place marker in cup", (2) "push button hard", (3) "pick red cube", (4) "place green cube in red bowl", and (5) "push red bowl to red cup". Tasks (1) and (2) are considered new tasks requiring 200 training samples due to their difficulty based on action complexity, physical demands (e.g., manipulating small objects like markers), and unique task specifications such as pressing the button hard in a particular demonstrated manner. Tasks (3)-(5) are in-domain tasks that require only 20 training samples. Importantly, all experiments use only 20 few-shot expert demonstrations for head selection when applicable, regardless of the total training data available. All models are finetuned for 5,000 iterations as detailed in Section 4.1 (implementation details). More details about the robot and the task setup can be found in Section C of the Supplement.

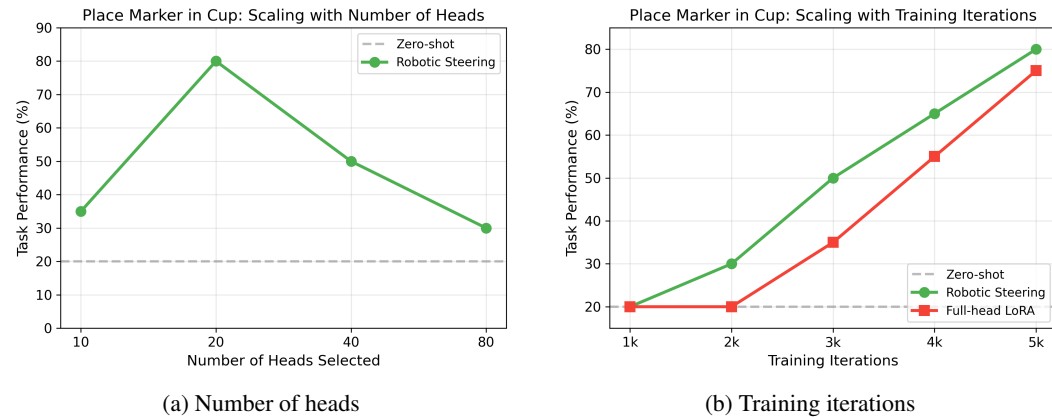

(a) Number of heads            (b) Training iterations

Figure 3: Scaling experiments on *Place Marker in Cup* task. (a) Success rate versus number of selected attention heads. (b) Success rate versus training iterations for Robotic Steering and Full-head LoRA, both starting from the same zero-shot baseline

## 5 RESULTS

Our main results are shown in Table 1. The crucial insight of Robotic Steering is that few-shot expert demonstrations can encode the physical nuances of robotic tasks and more importantly inform which task-specific components of a model to finetune.

Indeed, our results demonstrate that Robotic Steering matches or outperforms LoRA's success rate on all evaluated tasks. This is true for both simpler in-domain tasks which are similar to DROID [36] dataset tasks and excitingly more challenging, new tasks which we even provide 200 examples for finetuning. This demonstrates that Robotic Steering is a broadly effective finetuning approach, leveraging just 20 demonstrations to surpass the LoRA baseline. It is worth noting that none of these tasks are trivial given the physical context of on-robot evaluation as we see that zero-shot performance is near 0% success rate for all tasks. While $\pi_0$ is a SoTA VLA, it is remains brittle to generalization when faced with variations in environment conditions, robot embodiments, and language instructions. Beyond task performance, Table 1 demonstrates that Robotic Steering is significantly more computationally efficient than full-head LoRA, reducing finetuning time by 21% while using 96% fewer parameters. This efficiency is crucial for practical robotics, where rapid iteration and experimentation in new environments is essential. We present additional results and ablations in Section A of the Appendix.

### 5.1 ABLATIONS

We perform a comprehensive ablation study of Robotic Steering on the *Place Marker in Cup* task to understand the impact of key design choices. For all ablations, we use the base $\pi$-0 model.

**Varying number of attention heads**. In Figure 3a (a), we examine the impact of varying the number of selected attention heads used in our method. We find that performance peaks at 20 heads (80% success rate) and decreases with both fewer and more heads. This suggests that an optimal subset of heads exists for task-specific adaptation, where too few heads lack sufficient representational capacity while too many heads introduce noise or conflicting signals.

**Scaling with training iterations**. We investigate how our method scales with the number of training iterations compared to Full-head LoRA. As shown in Figure 3b, Robotic Steering demonstrates faster initial learning and achieves higher final performance (80%) compared to Full-head LoRA (75%) after 5k iterations. This result suggests that our approach scales well, surpassing or at least matching LoRA's capabilities of scaling performance with further training.

**Head selection approach**. Our results in Table 3 show that K-NN regression, our approach for head selection, slightly outperforms Causal Mediation Analysis (CMA) [63] and REINFORCE [30, 33]. CMA, specifically causal ablation in our experiments, selects heads by adding noise to each head

Table 2: Generalization performance under environmental variations and transfer to related tasks after training on Place Marker in Cup. ↓ indicates performance drop from base condition.

| Method | Base Performance | + Lighting Distractor | + Positional Variation | Unseen Task: Pick Mug | Unseen Task: Pl. Cube in Bowl |
|---|---|---|---|---|---|
| Zero-shot | 10% | 0% | 0% | 0% | 0% |
| Full-head LoRA | 75% | 40% ↓47% | 45% ↓X% | 0% | 40% |
| **Robotic Steering** | **80%** | **60%** ↓25% | **55%** ↓X% | **30%** | **55%** |

Table 3: Ablation studies on head selection methods and training components. All methods use 20 few-shot examples and select top-20 heads for adaptation.

| Method | Place Marker in Cup | Head Selection Time (min) | Fine-tuning Time (min) |
|---|---|---|---|
| *Head Selection Methods* | | | |
| CMA | 15% | 58 min | 51 min |
| REINFORCE | 80% | 93 min | 51 min |
| **K-NN Regression (Ours)** | **80%** | **17 min** | 51 min |
| *Training Components* | | | |
| Queries only | 10% | 17 min | 52 min |
| **Queries + MLP (Ours)** | **80%** | **17 min** | **51 min** |

and measuring the resulting performance drop on the 20 few-shot demonstrations. REINFORCE optimizes head selection through gradient-based search to maximize task performance. While all three methods achieve comparable task success rates as shown in Table 3, K-NN regression offers a crucial advantage: significantly lower runtime. This is due to K-NN regression not requiring model inference and evaluation for head selection. Once the activations are computed, K-NN regression boils down to a simple and very efficient retrieval-based regression on the activations themselves.

**Training Components**. We also carefully ablate the recipe for which precise components of the model to finetune. Of course, when selecting heads, it is natural to finetune their queries, but we also question whether additionally finetuning their MLPs, yields any benefit. Our results in Table 3 suggest that indeed finetuning both the queries and MLPs associated with the selected task-specific heads yields improvements in success rate. This suggests that the feedforward projection following attention is important to adapt for VLA finetuning. We do not consider finetuning the parameters of the keys and values as they are shared per layer in $\pi_0$'s base LLM [10].

## 5.2 ADDITIONAL EXPERIMENTS

In this subsection, we present experiments that demonstrate additional properties and capabilities of Robotic Steering, beyond its use for improving task-specific performance. Additional visualizations can be found in Supplementary Section A.2.2. For all experiments, we use the $\pi_0$ model with our steering method trained on *Place Marker in Cup*.

**Robustness to environmental distractors**. We evaluate the robustness and generalization of our method to common environmental variations that occur in real-world robotic deployment. We test our model trained on the base *Place Marker in Cup* task under two challenging conditions: lighting variations and positional distractors. As shown in Table 2, Robotic Steering shows significantly less degradation in the face of environmental variations. This demonstrates that our sparse head selection naturally filters out features sensitive to task-irrelevant variations while preserving task-critical attention heads.

**Zero-shot transfer to related tasks**. A key advantage of our approach is the ability to transfer learned steering vectors to related manipulation tasks without additional training. In Table 2, we evaluate the heads selected for *Place Marker in Cup* on two related tasks: *Pick Mug* and *Place Cube in Bowl*. Despite being trained only on the marker placement task, our method achieves 30% success

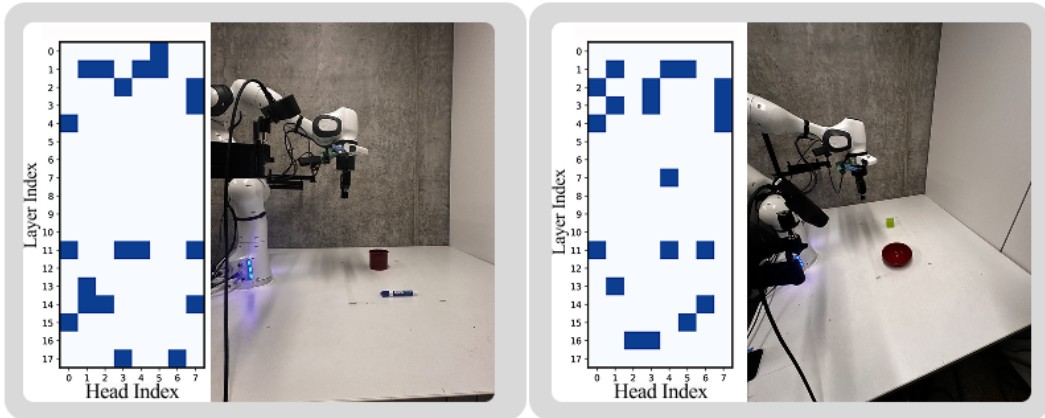

Figure 4: **Task and Head Selection Visual Left**: Attention heads selected by **Robotic Steering** and task visual for *Place Marker in Mug*; **Right:** Attention heads selected by **Robotic Steering** and task visual for *Place Cube in Bowl*

on *Pick Mug*, while Full-head LoRA completely fails (0%). This suggests that the sparse selected heads capture generalizable representations that transfer across tasks with similar action spaces and object interactions.

**Visualizing selected attention heads**. To understand what our method learns, we visualize the attention patterns of the top-selected heads for *Place Marker in Cup*, *Pick Cube*, and *Push Bowl to Cup* tasks in Figure 4. The visualizations reveal that different tasks activate distinct sets of heads. Intuitively, this aligns with the notion of functional specificity, except in models' attention heads. This interpretability is a key advantage of our approach: unlike black-box finetuning methods, we can directly inspect which attention mechanisms are being leveraged for each task.

## 6 CONCLUSION

In this work, we introduce Robotic Steering, which demonstrates that few-shot demonstrations can specify physically-grounded embodied tasks and help identify which specific attention heads in VLAs encode task-relevant physical reasoning. By selectively finetuning only these heads, we match or exceed full LoRA performance while using 96% fewer parameters and achieving superior robustness to environmental variations. Our visualizations reveal that different manipulation tasks activate distinct attention patterns, providing mechanistic insight into how VLAs encode physical tasks.

This work opens exciting research directions at the intersection of mechanistic interpretability and robotic learning. Future methods could explore alternative head selection approaches beyond K-NN regression, investigate finer-grained selection at the parameter or neuron level, or develop compositional schemes where multiple task-specific adaptations combine without interference. More fundamentally, our results suggest that the question of "what to finetune" deserves equal attention to "how to finetune", a shift that could transform how we adapt foundation models for robotics. As VLAs scale to billions of parameters, the ability to precisely identify and modify task-relevant components will become essential for practical deployment across the wide variety of physical contexts robots must master.

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
