# OpenReview forum: "Robotic Steering: Mechanistic Finetuning for Vision-Language-Action Models"
_ICLR.cc/2026/Conference — ICLR 2026 Conference Withdrawn Submission_

### Official Review · Reviewer_afF7 · 2025-10-17

**Soundness:** 2
**Presentation:** 1
**Contribution:** 2
**Rating:** 0
**Confidence:** 4

**Summary:**

Robotic Steering is the proposed approach to identify (from a few demonstrations) and fine-tune only task-specific attention heads in VLA models for robusteness to deployment in new settings. The authors establish training time as well as performance gains when compared against full head low rank adaptation (LoRA) fine-tuning.

**Strengths:**

The problem statement is very interesting and timely, as this problem of fine-tuning VLA models is of great interest to the community.

The solution proposed at a high level is elegant and simple enough to be appreciated as such.

The work is documented extensively in the sense of literature review and references to the state of the art.

The results presented appear to back the case the authors make.

**Weaknesses:**

**The work falls short of submission readiness:**
- Manuscript writing has not been completed, there are numerous missing references with empty brackets (lines 299, 303, 311) the appendix (which is inconveniently provided as a separate document) contains a whole section of empty placeholders (C ROBOTIC SETUP ADDITIONAL DETAILS)
- The ratio of useful information to figure size is extremely low (for all of figures 1,2,3,4). They are also of low quality. These appear, from a reviewer's point of view, to have been artificially bloated to occupy more space as to fill in room for work that is not of sufficient maturity. More specifically, figure 1 could be included in the text or as a table, the useful information there is about 20 words yet it takes 1/4 of a page with a redundant caption. It could also be combined with figure 2 that takes 2/3 of a page to basically explain the same thing with very poor design. The left plot in Figure 3 provides 4 data points + the baseline. Figure 4 is of extremely low visual quality and while authors claim (line 457) "The visualizations reveal that different tasks activate distinct sets of heads." it appears that there is a lot of overlap between the 2 tasks' activation regions.
- The results section is largely breadth with no depth of exploration or analysis of any of the experiments listed. The statistical significance of success rates provided in table 1 are questionable over only 20 rollouts, especially when information on the scoring protocol/failure cases is not provided.
- Furthermore, it is also full of vacuous conclusions such as (lines 458-460) "This interpretability is a key advantage of our approach: unlike black-box finetuning methods, we can directly inspect which attention mechanisms are being leveraged for each task." what does this mean? how is it useful? why is this an interesting result given that by design you have selected those top heads in the first place?

**Core contribution is not clearly motivated/presented:**
- The intuition behind the k-NN regression technique is not very clear (lines 222-225) and warrants further discussion and illustration. Also, decoupling trajectories and using per-timestep examples only is done without any explanation or justification and it is unclear why this would be the preferred approach.

I will limit my exposé to these points as I urge the authors to heavily revise the presentation of the manuscript and complete the work. This should also include a reproducibility statement as well as a statement on the use of LLMs.

**Questions:**

- If fine-tuning is done on a specific task, and it appears some heads share influence for multiple tasks, what happens to the success rate of task B if I fine-tune on task A.

- Can you explain the task performance vs number of heads selected plot (figure 3). Shouldn't increasing the number of heads lead us to the full head LoRA case which performs at a similar level as with 20 selected in this case? Why the drop-off?? Again the explanation given in the paper (lines 368-370) is not of top conference level quality and does not provide any insight into the mechanisms behind this drop in performance.

---

### Official Review · Reviewer_zqnF · 2025-10-20

**Soundness:** 4
**Presentation:** 3
**Contribution:** 3
**Rating:** 6
**Confidence:** 4

**Summary:**

The authors introduce 'Robotic Steering,' a LoRA fine-tuning methodology for VLAs. Their approach leverages few-shot demonstrations of the robotic task to identify a sparse subset of attention heads which are most relevant for the given robotic task; this step is performed using kNN regression on attention head activations which are most predictive of the ground truth actions. The heads selected by kNN are subsequently fine-tuned using LoRA, rather than using LoRA uniformly across all attention layers. The authors provide empirical evidence that their approach matches or exceeds traditional LoRA fine-tuning but with less parameters.

**Strengths:**

Application of mechanistic interpretability (MI): this paper proposes a creative and effective technique for VLA fine-tuning. While MI techniques are gaining traction in natural language processing, their application in robotics is nascent. This paper provides a clear and effective framework for borrowing tools from that field and applying them to robotics.

Real-world evals: the claims in the paper are supported by real-world experiments on the Franka robot. The results in Table 1 clearly demonstrate Robotic Steering decreases computational cost (training time and parameter account), assuming one has N expert demonstrations and has already completed the kNN step (see comment below).

Ablations: the authors conducted numerous ablations which are summarized in Figure 3; this helps to contextualize the strengths and weaknesses of their method, and is a valuable addition to the paper.

Robustness: as depicted by Table 2, Robotic Steering is possibly more robust to environmental variations than full LoRA fine-tuning. This is important for adapting VLAs to real-world tasks, and this provides preliminary evidence of a more effective fine-tuning strategy.

**Weaknesses:**

Unaddressed computational cost of attention head selection: the paper highlights the efficiency gains in training after performing the mechanistic interpretability step, which often requires the most memory and run-time due to the need to cache activations. A complete picture of the method's efficiency would include a discussion of resource requirements of this step.

Hyperparameter sensitivity: the ablation in Figure 3 reveals performance is very sensitive to the number of selected heads, which appears task and model specific. This adds and extra layer of complexity to the fine-tuning process, which may not warrant use over traditional LoRA fine-tuning methods, which don't require a thorough hyperparameter search.

Generality vs. Task-Specificity: Robotic Steering is task-specific, which is where a lot of its benefits come from. However, this raises questions about scalability when a robot must be adapted to perform a suite of tasks. The paper would be strengthened by a discussion of this trade-off and potential work to address it.

**Questions:**

Clarification of Training Time: does the training time in Table 1 include the kNN procedure, or just training alone? Can you provide a breakdown of the full time cost of Robotic Steering?

Memory requirements: can you please specify the memory requirements in performing the identification of task-relevant attention heads (for use in kNN)? Specifically, how large are the activation caches over the few-shot demos? How does this scale with the number of demos and the size of the base VLA?

Incorrect citation: the citation for Jax in Section 4.1 is wrong and the relative changes in performance in Table 2 are wrong (only placeholders).

---

### Official Review · Reviewer_BXNE · 2025-10-26

**Soundness:** 3
**Presentation:** 4
**Contribution:** 4
**Rating:** 6
**Confidence:** 3

**Summary:**

The paper “Robotic Steering: Mechanistic Finetuning of Vision-Language-Action Models” proposes a new fine-tuning paradigm for Vision-Language-Action (VLA) models inspired by mechanistic interpretability. Unlike traditional parameter-efficient methods (e.g., LoRA) that adapt all or fixed subsets of weights, Robotic Steering selectively fine-tunes only those attention heads that are task-relevant — identified via few-shot expert demonstrations. The method proceeds in three steps: (1) Semantic Attribution: Identifies attention heads whose activations best predict ground-truth actions across demonstration trajectories using a k-NN regression score. (2) Selective Finetuning: Applies LoRA adapters only to query and MLP weights of the selected heads. (3) Standard Inference: The finetuned model runs as a normal checkpoint without any runtime overhead.

On-robot evaluations with a Franka Emika arm show that Robotic Steering matches or exceeds full-head LoRA performance across multiple tasks such as “place marker in cup” and “push button,” while using 96% fewer trainable parameters and achieving faster convergence (21% less time). Moreover, it exhibits better robustness to lighting, object, and positional variations and allows zero-shot transfer of selected heads to related tasks.

**Strengths:**

- The paper is among the first to operationalize mechanistic interpretability for real robotic control. By identifying and fine-tuning task-relevant attention heads, the authors propose a method that is both interpretable and effective—something rare in robotic finetuning literature.

- Results on the Franka Emika arm, rather than simulation alone, make this paper stand out. The tasks are physically diverse and challenging, including small-object manipulation and spatial reasoning under real lighting and camera constraints.

- Robotic Steering achieves the same or higher success rates than full-head LoRA while using less than 5% of the parameters. This not only makes it computationally efficient but also empirically demonstrates improved robustness to environmental changes and transfer to unseen but related tasks.

- The visual analysis of head activations across different tasks provides genuine mechanistic insight, revealing that distinct heads correspond to distinct physical behaviors (e.g., grasping vs. pushing). This interpretability is a strong practical advantage over black-box finetuning.

- The paper rigorously tests the method under varying head counts, selection methods (KNN, CMA, REINFORCE), and fine-tuned components (queries only vs. queries+MLPs). These experiments convincingly show that the chosen design choices are optimal.

**Weaknesses:**

- While the approach draws analogies to neuroscience and functional specialization, the paper does not provide a formal justification or quantitative metric for “task-relevance” beyond empirical correlation with k-NN action prediction. A more principled interpretation (e.g., mutual information between activations and action space) would strengthen the claim.

- The approach assumes high-quality, representative few-shot demos to identify relevant heads. If demonstrations are noisy or incomplete, the head attribution process might select spurious components, potentially hurting performance.

- The zero-shot transfer experiments are limited to closely related tasks (e.g., cup → bowl). It remains uncertain whether the same mechanism scales to semantically distinct or multi-step tasks.

- While efficient for modest-sized VLAs (π₀), scaling to larger multimodal backbones (e.g., Octo, PaLI-X) might make per-head activation analysis computationally intensive. The paper would benefit from a brief discussion on scaling implications.

- Works on task vectors or activation steering in multimodal settings (e.g., Sparse Attention Vectors, Function Vectors) could have been empirically compared, even qualitatively, to contextualize where mechanistic finetuning excels or differs.

**Questions:**

- How sensitive is the performance to the number of selected heads (m)? Does over-selecting degrade performance primarily due to noise or conflicting representations?

- Could head selection be made adaptive across time rather than static before finetuning?

- Is the identified head subset consistent across random seeds or different demonstration sets for the same task?

- How does Robotic Steering perform if used in conjunction with in-context or retrieval-based adaptation rather than as a replacement?

- Could the approach be extended to dynamically re-steer during deployment based on new sensory input (online adaptation)?

---

### Official Review · Reviewer_Ngkg · 2025-11-01

**Soundness:** 1
**Presentation:** 2
**Contribution:** 1
**Rating:** 2
**Confidence:** 3

**Summary:**

The authors introduce Robotic Steering, a parameter-efficient fine-tuning method for VLAs that focuses on increasing the efficiency of fine-tuning VLAs on demonstration data for real-world deployment. Robotic steering is driven by the intuition that task-specific information is encoded in different attention heads.  Demonstration examples of a task, typically collected in the target environment for fine-tuning, are passed through the VLA. The attention heads are grouped according to cosine similarity of the activations, designated as task-relevant heads and fine-tuned with LoRA. The experiments are run on a real-world robot set-up similar to DROID where robotic steering is shown to outperform LoRA fine-tuning of all parameters.

**Strengths:**

- Method is clearly motivated and interpretable
- Results on real robot set up are promising

**Weaknesses:**

- Incomplete exploration of novel idea: the idea of aggregating the attention heads of a VLA based on their activations on few-shot demonstrations and using those to prune the parameters fine-tuned is novel, but needs further exploration. See questions below.
- Experiments:
    - Full LoRA fine-tuning is the only baseline included. Other PEFT methods such as ECoT [1] should be considered
    - Only one environment is considered. It’s unclear whether the results transfer across environments.
    - Only one base VLA model is considered ($pi_{0}$) that depends on a diffusion loss. It’s unclear whether the results transfer across different architectures of VLA models, such as those that use Next Token Prediction or Cross Entropy loss, which is one of the claims of the paper (l. 298 “model-agnostic”).

**Questions:**

l. 88-89 Are you implying that there are parameters that are not helpful to fine-tune during full fine-tuning? One could test this by measuring the magnitude of the gradient updates to the non-task-specific attention heads during full fine-tuning.

Section 3.2: How do you ensure that you’re selecting activations associated with nonzero/meaningful actions?

Section 4.2: What was the model pretrained on? Why are Tasks 1-2 considered OOD but Tasks 3-5 are in-domain?

Table 1: Why didn’t you include full fine-tuning of $pi_{0}$ (LoRA and non-LoRA) as baselines, to represent the upper limit of performance?

Figure 3: a) Shouldn’t Robotic Steering’s performance approach full-head LoRA’s as the number of selected heads increases? It drops way below 75%. b) It seems the trend continues, why did you stop at 5k iterations?

What happens if you combine the weights from the separately trained heads?

[1] “Robotic Control via Embodied Chain-of-Thought Reasoning.” Zawalski et al., arxiv preprint arXiv:2407.08693. 2024.

---

### Note · Authors · 2025-11-14

**Comment:**

Re-submitted

**Withdrawal Confirmation:**

I have read and agree with the venue's withdrawal policy on behalf of myself and my co-authors.